# Selective production of methylindan and tetralin with xylose or hemicellulose

Zhufan Zou[1,2,3,6], Zhenjie Yu[1,3,6], Weixiang Guan[1], Yanfang Liu[4], Yumin Yao[4], Yang Han[4], Guangyi Li[1], Aiqin Wang [1,5], Yu Cong [1], Xinmiao Liang[4] ✉, Tao Zhang [1,2,5] ✉ & Ning Li [1] ✉

Indan and tetralin are widely used as fuel additives and the intermediates in the manufacture of thermal-stable jet fuel, many chemicals, medicines, and shockproof agents for rubber industry. Herein, we disclose a two-step route to selectively produce 5-methyl-2,3-dihydro-1*H*-indene (abbreviated as methylindan) and tetralin with xylose or the hemicelluloses from agricultural or forestry waste. Firstly, cyclopentanone (CPO) was selectively formed with ~60% carbon yield by the direct hydrogenolysis of xylose or hemicelluloses on a non-noble bimetallic Cu-La/SBA-15 catalyst. Subsequently, methylindan and tetralin were selectively produced with CPO via a cascade self-aldol condensation/rearrangement/aromatization reaction catalyzed by a commercial H-ZSM-5 zeolite. When we used cyclohexanone (another lignocellulosic cycloketone) in the second step, the main product switched to dimethyltetralin. This work gives insights into the selective production of bicyclic aromatics with lignocellulose.

For sustainable development and environment protection, the catalytic transformation of biomass to fuels[1–13] and chemicals[14–22] has aroused broad concern. Aromatics are important fuel components and intermediates in the manufacture of many useful chemicals. During the past decades, great efforts have been devoted to the selective synthesis of monocyclic aromatics (such as benzene, toluene and xylene) with lignocellulosic platform compounds[23–28]. In contrast, few work has been done about the selective synthesis of more valuable bicyclic aromatics. Indan and tetralin are two important jet fuel additives[29–32]. After being hydrogenated, they can be converted to octahydroindene and decalin, two major components of the famous thermal-stable aviation fuel JP-900[32]. Meanwhile, these compounds are also widely utilized as intermediates in the manufacture of many chemicals, medicines or shockproof agent in rubber industry[33–37]. Currently, indan and tetralin are obtained from nonrenewable fossil energy at low yields. From the pointview of sustainable development, it is still necessary to explore new method for the production of indan and tetralin with renewable,

cheap and widely available biomass. Hemicellulose is one of the most important components of biomass. In practical, hemicellulose can be readily separated from forest residues and agriculture wastes with dilute acid or hot water[38–40]. Therefore, the production of indan (or alkylated indan) and tetralin with hemicellulose (or xylose from the hydrolysis of hemicellulose) is of great significance.

Herein, we reported a two-step method for the synthesis of renewable methylindan and tetralin by the direct hydrogenolysis of xylose (or hemicellulose) to CPO on a bimetallic Cu-La/SBA-15 catalyst, followed by a cascade self-aldol condensation/rearrangement/aromatization of CPO over a commercial H-ZSM-5 zeolite. The strategy was shown in Supplementary Fig. 1.

## Results and discussion

### Hydrogenolysis of xylose or hemicellulose to CPO

CPO is a widely used intermediate in the manufacture of perfume, medicine, pesticides, synthetic rubber, etc.[41]. In our very recent work[42],

[1]CAS Key Laboratory of Science and Technology on Applied Catalysis, Dalian Institute of Chemical Physics, Chinese Academy of Sciences, Dalian, China. [2]School of Chemistry, Dalian University of Technology, Dalian, China. [3]University of Chinese Academy of Sciences, Beijing, China. [4]Key Lab of Separation Science for Analytical Chemistry, Dalian Institute of Chemical Physics, Chinese Academy of Sciences, Dalian, China. [5]State Key Laboratory of Catalysis, Dalian Institute of Chemical Physics, Chinese Academy of Sciences, Dalian, China. [6]These authors contributed equally: Zhufan Zou, Zhenjie Yu. ✉e-mail: liangxm@dicp.ac.cn; taozhang@dicp.ac.cn; lining@dicp.ac.cn

it was found for the first time that CPO could be directly obtained from the selective hydrogenolysis of xylose and hemicellulose over Ru/C catalyst. Taking into consideration the high price and low reserves of Ru, it is imperative to explore the non-noble metal catalysts that are effective for the direct synthesis of CPO from xylose or hemicellulose. In the first part of this work, we explored the hydrogenolysis of xylose over the SBA-15 loaded non-noble metal catalysts (denoted as M/SBA-15, M = Cu, Co, Ni) in toluene/NaCl biphasic reaction system. Based on Fig. 1 and Supplementary Figs. 2-5, significantly higher CPO carbon yield was reached over the Cu/SBA-15 catalyst. The presences of chloride salts (such as NaCl and KCl) and organic solvents were also found to be favorable for the conversion of xylose to CPO on the Cu/SBA-15 catalyst, especially when the reaction was performed in a biphasic system of toluene and 3 wt% NaCl aqueous solution. The promotion effect of NaCl can be comprehended by two reasons: (1) From Fig. 1a and Supplementary Fig. 6, we can see that furfural was the main product when we conducted the reaction without using any metallic catalyst. According to the analysis of the products that were obtained in the biphasic reaction systems of toluene/NaCl aqueous solution and tolene/$H_2O$ in short reaction time (35 min), the presence of NaCl promoted the generation xylulose and furfural (Supplementary Figs. 7 and 8). On the basis of literature[43,44] and our recent work[36], this result can be rationalized because chloride salts can act as Lewis acid to accelerate the isomerization of xylose to xylulose that is easier to be dehydrated to furfural than xylose. Subsequently, furfural can be further converted CPO by the aqueous phase selective hydrogenolysis over Cu-based catalyst[45–47]. (2) According to Fig. 1d, the presence of

NaCl also restrains the hydrogenation of xylose and furfural to xylitol and tetrahydrofurfuryl alcohol (THFA) over the Cu/SBA-15 catalyst, which can be considered as another reason for the promotional effect of NaCl. To verify this, we studied the hydrogenolysis of xylitol and THFA under the investigated conditions, respectively. As we expected, the conversions of xylitol (3.2%) and THFA (5.1%) over the Cu/SBA-15 catalyst were very low, no CPO (or cyclopentanol) was detected in the products. Analogously, the addition of organic solvent decreases the furfural concentration in aqueous phase. From the pointviews of reaction equilibrium and restraining the formation of humins, this is favorable for the transformation of xylose to furfural[43,48,49]. After the optimization of xylose concentration and reaction temperature, high yield (~ 60%) of CPO was reached over the Cu/SBA-15 catalyst. As far as we know, this should be the first report on the direct production of CPO from xylose over non-noble metal catalyst.

For real application, we studied the reusability of Cu/SBA-15. According to Fig. 2, the activity of Cu/SBA-15 catalyst for the hydrogenolysis of xylose to CPO significantly decreased after usage under the optimized reaction conditions (Fig. 2a). After being doped with small amounts (3 wt%) of rare-earth metals (such as La, Ce, Y), the initial activity and selectivity of Cu/SBA-15 in the hydrogenolysis of xylose to CPO didn't change too much, while the reusability of Cu/SBA-15 catalyst was greatly improved (Figs. 2a, 2b, 2d–f and Supplementary Fig. 9 (the results obtained at lower xylose conversion)). Although slightly deactivation still could be noticed during the repeated usage of the bimetallic catalysts, such a problem can be overcome by the regeneration of the used bimetallic catalysts by the calcination in air

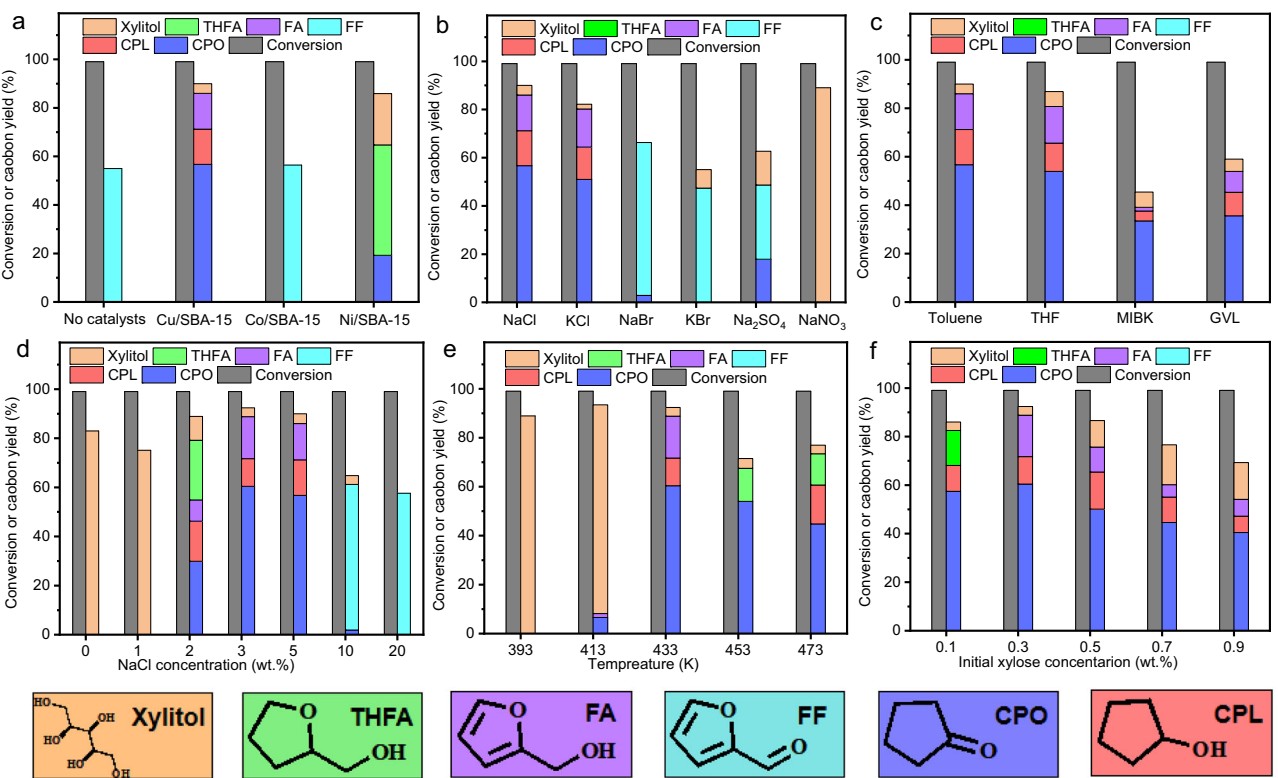

**Fig. 1 | Catalytic conversion of xylose on the non-noble metal catalysts.** Xylose conversion and the carbon yields of various products on the M/SBA-15 (M = Cu, Co, Ni) catalysts **a**. Xylose conversion and the carbon yields of products on the Cu/SBA-15 catalyst as the functions of salt type **b**, solvent **c**, salt concentration **d**, reaction temperature **e**, and initial xylose concentration **f**. Reaction conditions: **a** 433 K, 3 MPa $H_2$, 500 rpm, 4 h; 10 mL 5 wt.% NaCl aqueous solution, 10 mL toluene, 0 or 30 mg M/SBA-15 catalyst and 0.3 g xylose were utilized for the tests. **b** 433 K, 3 MPa $H_2$, 500 rpm, 4 h; 10 mL 5 wt.% salts aqueous solution, 10 mL toluene, 0.3 g xylose and 30 mg Cu/SBA-15 were used for the tests. **c** 433 K, 3 MPa $H_2$, 500 rpm, 4 h; 10 mL

5 wt.% NaCl aqueous solution, 10 mL organic solvent, 0.3 g xylose and 30 mg Cu/SBA-15 were used for the tests. **d** 433 K, 3 MPa $H_2$, 500 rpm, 4 h, 10 mL NaCl aqueous solution, 10 mL toluene, 0.3 g xylose and 30 mg Cu/SBA-15 were used for the tests. **e** 3 MPa $H_2$, 500 rpm, 4 h; 10 mL 3 wt% NaCl aqueous solution, 10 mL toluene, 0.3 g xylose and 30 mg Cu/SBA-15 were utilized for the tests. **f** 433 K, 3 MPa $H_2$, 500 rpm, 4 h, 10 mL 3 wt.% NaCl aqueous solution, 10 mL toluene, xylose and Cu/SBA-15 (Cu/SBA-15 to xylose mass ratio = 1/10) were used for the tests. THFA tetrahydrofurfuryl alcohol. FA furfuryl alcohol. FF furfural, CPO cyclopentanone, CPL cyclopentanol.

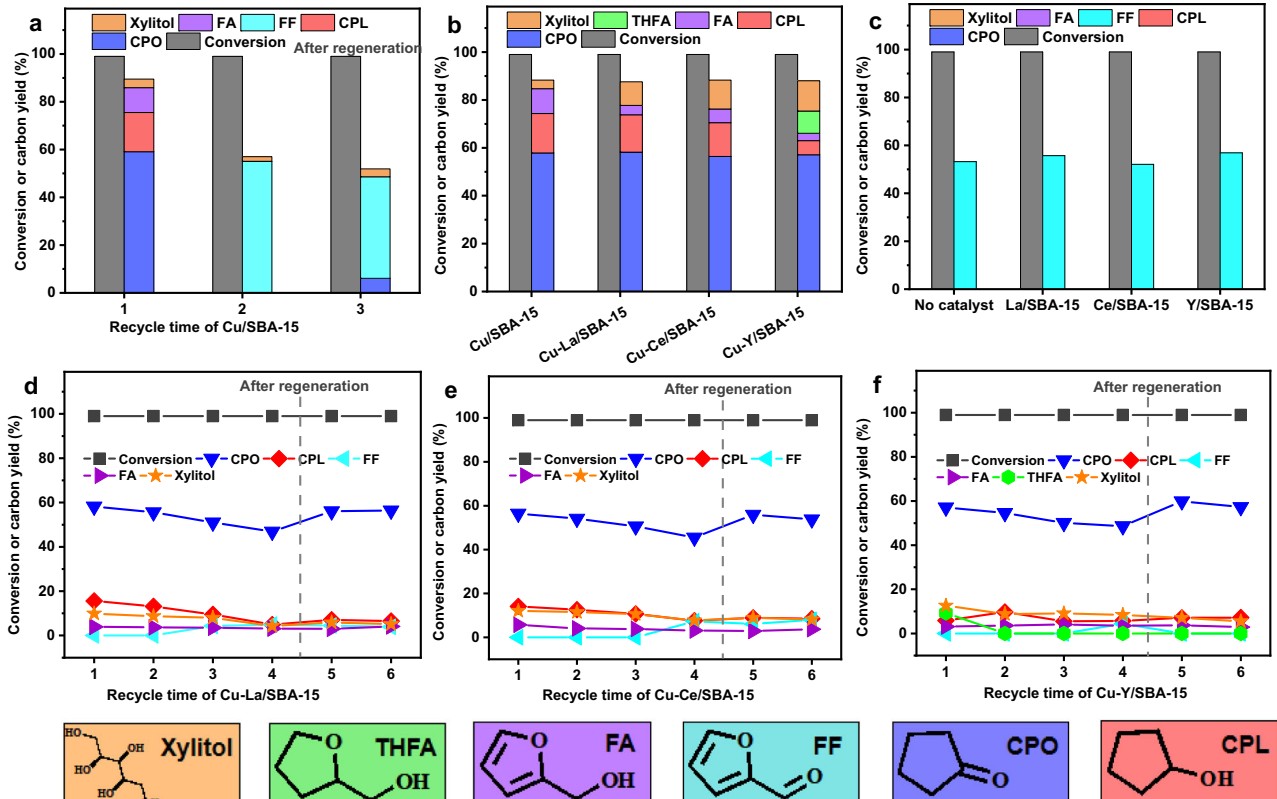

**Fig. 2 | Catalytic conversion of xylose over the Cu/SBA-15, X/SBA-15 and Cu-X/SBA-15 (X = La, Ce, Y) catalysts.** Reusability of Cu/SBA-15 catalyst **a** and Cu-X/SBA-15 **d**–**f** catalysts. Conversion of xylose and the carbon yields of products over the Cu-X/SBA-15 (**b**) and X/SBA-15 (**c**) catalysts. Reaction conditions: 433 K, 3 MPa H₂, 500 rpm, 4 h; 10 mL 3 wt% NaCl aqueous solution, 10 mL toluene, 0.3 g xylose and 30 mg catalyst were utilized for the test. The contents of rare-earth metal in Cu-La/ SBA-15, Cu-Ce/SBA-15, Cu-Y/SBA-15, La/SBA-15, Ce/SBA-15 and Y/SBA-15 catalysts were fixed as 3 wt%. In the regeneration process, the used catalysts were calcinated in air flow at 773 K for 4 h and reduced in 20% H₂/Ar flow at 773 K for 4 h. THFA tetrahydrofurfuryl alcohol. FA furfuryl alcohol, FF furfural, CPO cyclopentanone, CPL cyclopentanol.

flow at 773 K for 4 h and the reduction in 20% H₂/Ar flow at 773 K for 4 h. From Fig. 2c, we can see that the La/SBA-15, Ce/SBA-15 and Y/SBA-15 catalysts are inactive for the hydrogenolysis of xylose to CPO. Therefore, we can't simply attribute the higher reusability of bimetallic catalysts to the presence of rare-earth metals. Based on the TG-DTA analysis (Supplementary Fig. 10), the deactivation of the bimetallic catalysts may be rationalized by the formation of coke by the polymerization of furfuryl alcohol (an intermediate that may generated during the conversion of xylose to CPO)[45,46,50]. This should be the reason why the activity of these catalysts can be restored by the calcination in flowing air followed by the reduction by 20% H₂/Ar. Among the investigated bimetallic catalysts, Cu-La/SBA-15 has the highest reusability, which is advantageous in future application.

To figure out the reason for the stabilizing effect of La on Cu/ SBA-15, we characterized the catalysts by ICP and TEM (Supplementary Table 1 and Supplementary Fig. 11). It was found that the modification of Cu/SBA-15 by small amount of La significantly restrained the leaching of Cu species and the sintering of Cu particles, which may be the reasons for the higher stability of the Cu-La/SBA-15 catalyst in the hydrogenolysis of xylose to CPO. To get deeper insight of the influences of NaCl on the Cu-La/SBA-15 catalyst, we compared the catalytic performances of the fresh and NaCl treated Cu-La/SBA-15 catalysts in the hydrogenation furfural (the main product from the conversion of xylose in the absence of any metallic catalyst). Based on the results illustrated in Supplementary Fig. 12, furfural was completely hydrogenated to tetrahydrofurfuryl alcohol over the fresh Cu-La/SBA-15 catalyst, while a mixture of furfuryl alcohol and tetrahydrofurfuryl alcohol was obtained over the NaCl treated Cu-La/ SBA-15 catalyst. When the hydrogenation of furfural was carried out

in a toluene/H₂O biphasic reaction system, CPO was obtained as the main product over the NaCl treated Cu-La/SBA-15 catalyst. Based on these results, we can draw two conclusions: (1) the presence of Cl⁻ ion in the reaction system restrains the hydrogenation of furan ring over Cu-La/SBA-15, which may be the second reason for the promotion effect of NaCl; (2) the presence of water is necessary for the generation of CPO from the hydrogenation of furfural. It is worthy mention that the treatment of Cu-La/SBA-15 catalyst with NaCl also decreased its activity for the hydrogenation of xylose to xylitol (Supplementary Fig. 13), which should be the third reason for the promotion effect of NaCl. According to TEM-EDX elemental mappings and XPS results (Supplementary Figs. 14 and 15), the treatment of Cu-La/SBA-15 catalyst with NaCl aqueous solution (or usage in the toluene/NaCl aqueous solution biphasic reaction system) led to the partial coverage of metal particles by Cl species and the electron transfer from Cu species to Cl species, which may be the intrinsic reason for the restraining effects of NaCl.

From Fig. 2, we can see that cyclopentanol was also obtained as a by-product in the hydrogenolysis of xylose over Cu-La/SBA-15 catalyst. In real application, we can further increase the CPO carbon yield by dehydrogenation. To verify this hypothesis, we did an additional experiment using a commercial Pd/C catalyst. After the further dehydrogenation of the toluene phase product from the hydrogenolysis of xylose at 463 K for 4 h, the CPO carbon yield was improved from 58.5% to 65.8% (Supplementary Fig. 16). For environmental protection, we also checked the reuseability of the NaCl aqueous solution. To do this, we separated the aqueous phase product (a mixture of water, NaCl and small amount of sorbitol) by a separatory funnel and directly reused it as the NaCl aqueous solution in the next batch reaction. As we can see

from Supplementary Fig. 17, the NaCl aqueous solution can be repeatedly used without any treatment, which is advantageous in real application.

Moreover, we also explored the applicability of Cu-La/SBA-15 catalyst for the synthesis of CPO with other pentoses (such as arabinose, xylulose) or the hemicelluloses extracted from poplar wood or corncob (denoted as hemicellulose$_{poplar\ wood}$ and hemicellulose$_{corncob}$,

**Table 1 | Conversions of substrates and the carbon yields of various products over the Cu-La/SBA-15 catalyst.[a]**

| Substrate[b] | Conversion of substrate (%) | Carbon yields of different products (%)[c] | | | | | |
|---|---|---|---|---|---|---|---|
| | | MTHF | MF | CPO | CPL | THFA | Xylitol |
| Xylose | >99 | — | — | 58.2 | 15.6 | — | 3.9 |
| Xylulose | >99 | — | — | 60.4 | 11.2 | 3.3 | — |
| Arabinose | >99 | — | 3.9 | 55.1 | 9.7 | — | — |
| Hemicellulose$_{poplar\ wood}$ | >99 | 1.5 | — | 57.5 | 11.3 | 4.4 | — |
| Hemicellulose$_{corncob}$ | >99 | — | 2.1 | 54.9 | 8.5 | — | 3.6 |

[a] Reaction conditions: 433 K, 3 MPa H$_2$, 500 rpm, 4 h, 10 mL 3 wt% NaCl aqueous solution, 10 mL toluene, 0.3 g substrate, and 30 mg Cu-La/SBA-15 were used for the tests. For the extracted hemicelluloses, 10 mL hemicellulose aqueous solution, 10 mL toluene, 0.3 g NaCl and 30 mg Cu-La/SBA-15 were used for the tests.

[b] Hemicellulose$_{poplar\ wood}$ and hemicellulose$_{corncob}$ denote the hemicellulose solutions extracted from poplar wood or corncob. Based on Supplementary Table 2, the total concentrations of arabinose and xylose in the hemicellulose solutions were 1.1 wt%.

[c] MTHF methyltetrahydrofuran. MF methylfuran, CPO cyclopentanone, CPL cyclopentanol, THFA tetrahydrofurfuryl alcohol.

see Supplementary Table 2 for detail information). As we anticipated, ~60% CPO carbon yields were obtained under the similar conditions (Table 1). For industrial production, this is profitable because the substitution of xylose with the hemicellulose solutions extracted from agriculture and forest resides can greatly reduce the cost, equipment investment and energy consumption in the production of CPO.

## Synthesis of methylindan and tetralin with CPO

As the second innovation, we found that CPO could be directly converted to 5-methyl-2,3-dihydro-1*H*-indene (abbreviated as methylindan), tetralin and small amounts of their dehydrogenation products (i.e., methylindene and naphthalene) over a series of acidic zeolites (Fig. 3 and Supplementary Figs. 18-23). As far as we know, this should be the first quantitative report on the selective production of aromatics (especially methylindan and tetralin) by the reaction of biomass derived oxygenates over zeolite catalyst. Besides these C$_{10}$ aromatics, benzene, toluene, xylene, trimethylbenzene (denoted as C$_6$-C$_9$ aromatics) and C$_{11}$-C$_{12}$ bicyclic aromatics were also detected in the products. In the light of these results, a series of reaction pathways were proposed in Supplementary Fig. 24.

Among the investigated zeolite catalysts, the H-ZSM-5 demonstrated relatively higher activity and selectivity for the conversion of CPO to methylindan and tetralin (Fig. 3). The advantage of H-ZSM-5 was further verified by the additional experiments that were carried out over the zeolite catalysts with same amount of acid sites. To do this, we first measured the acid amounts of different zeolites by NH$_3$-chemisorption, then calculated the dosages of catalysts based on the NH$_3$-chemisorption results and conducted the activity tests

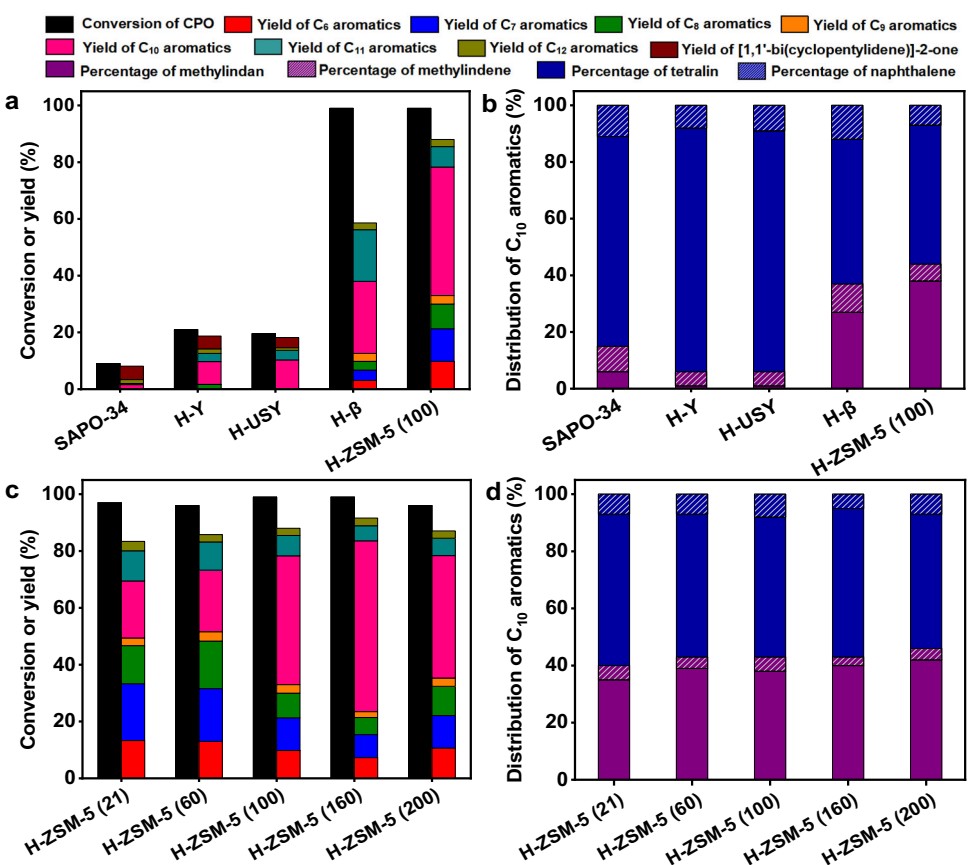

**Fig. 3 | Catalytic conversion of CPO over the zeolite catalysts. a** CPO conversion and the carbon yields of products over different zeolite catalysts. **b** Distributions of C$_{10}$ aromatics (including methylindan, methylindene, tetralin and naphthalene) over different zeolite catalysts. **c** CPO conversions and the carbon yields of products over the H-ZSM-5 catalysts with different SiO$_2$/Al$_2$O$_3$ molar ratios.

**d** Distribution of C$_{10}$ aromatics (including methylindan, methylindene, tetralin and naphthalene) over the H-ZSM-5 catalysts with different SiO$_2$/Al$_2$O$_3$ molar ratios. Reaction conditions: 723 K, 0.1 MPa N$_2$, WHSV = 0.45 g g$^{-1}$ h$^{-1}$, the initial N$_2$/CPO molar ratio = 36/1. CPO cyclopentanone, WHSV weight hour space velocity.

(Supplementary Table 3 and Supplementary Fig. 25). According to literature[51-53], the excellent performance of H-ZSM-5 may be comprehended by its special three-dimensional ten-ring channel structure that is suitable for the production of aromatics. Furthermore, we also studied the influence of $SiO_2/Al_2O_3$ molar ratio on the conversion of CPO over H-ZSM-5 catalyst. Based on Fig. 3, the total yield of $C_{10}$ aromatics (mainly composed of methylindan and tetralin) over H-ZSM-5 catalyst increased with the $SiO_2/Al_2O_3$ molar ratio, reached maximum when we used a H-ZSM-5 with a $SiO_2/Al_2O_3$ molar ratio of 160 (denoted as H-ZSM-5 (160)) as the catalyst, then decreased with the further increment of $SiO_2/Al_2O_3$ molar ratio. Over the H-ZSM-5 (160) catalyst, high total carbon yield (65%) of methylindan and tetralin was reached after the optimization of the reaction temperature and weight hour space velocity (WHSV) (Fig. 4). Based on the $N_2$-physisorption, $NH_3$-chemisorption, $NH_3$-TPD results and the FT-IR spectra using pyridine as probing molecule (Supplementary Table 4 and Supplementary Figs. 26 and 27), the good performance of the H-ZSM-5 (160) catalyst can be rationalized by its relatively higher specific BET surface area, higher strong acid sites/weak acid sites ratio and lower Brönsted acid sites/Lewis acid sites ratio.

It is worthy mention that [1,1'-bi(cyclopentylidene)]−2-one was also identified in the products that were obtained over the H-ZSM-5 catalyst at relatively low reaction temperature or high WHSV (Supplementary Figs. 28 and 29). According to literature[54], the [1,1'-bi(cyclopentylidene)]−2-one was generated from the self-aldol condensation of CPO, a reaction that can be catalyzed either by basic or acidic catalysts. Therefore, we think that [1,1'-bi(cyclopentylidene)]−2-one may be considered as an intermediate in the conversion of CPO to methylindan and tetralin. To verify this speculation, we studied the

conversion of [1,1'-bi(cyclopentylidene)]−2-one over H-ZSM-5 catalysts (Fig. 5). As we predicted, higher total carbon yields of methylindan and tetralin were reached under the same reaction conditions.

In the previous work Agrawal and Jones[55], a qualitative research based on the $^{13}C$ and $^{29}Si$ MAS NMR spectra was carried out about the conversion of CPO to bicyclic aromatics over H-Y zeolite. Due to the limitation of research method, they just pointed out the possibility to produce bicyclic aromatics with CPO. However, they didn't give the CPO conversions and the carbon yields of different products. In that work, it was suggested that the $C_{10}$ aromatics were generated by the further reaction of CPO trimer (generated from deep condensation of [1,1'-bi(cyclopentylidene)]−2-one and CPO). To check if this mechanism also applies to our reaction system, we conducted some additional experiments using CPO, [1,1'-bi(cyclopentylidene)]−2-one or their mixture (at the initial molar ratio of 1:1) as the feedstocks, respectively. According to Supplementary Fig. 30, higher $C_{10}$ bicyblic aromatics carbon yield (or selectivity) was achieved when we used [1,1'-bi(cyclopentylidene)]−2-one as the feedstock under the investigated conditions, which means [1,1'-bi(cyclopentylidene)]−2-one needn't to react with CPO before it was converted to $C_{10}$ bicyblic aromatics. To further confirm this speculation, we also compared the reactivity of CPO, [1,1'-bi(cyclopentylidene)]−2-one and their mixture at lower reaction temperature. Based on Supplementary Fig. 31, [1,1'-bi(cyclopentylidene)]−2-one exhibited higher reactivity than those of CPO and the mixture of [1,1'-bi(cyclopentylidene)]−2-one and CPO. Based on above results, we believe that [1,1'-bi(cyclopentylidene)]−2-one can be directly converted to $C_{10}$ aromatics without passing a trimerization step.

Moreover, we also investigated the conversion of [1,1'-bi(cyclo-pentylidene)]−2-one over the H-ZSM-5 (160) catalyst at lower

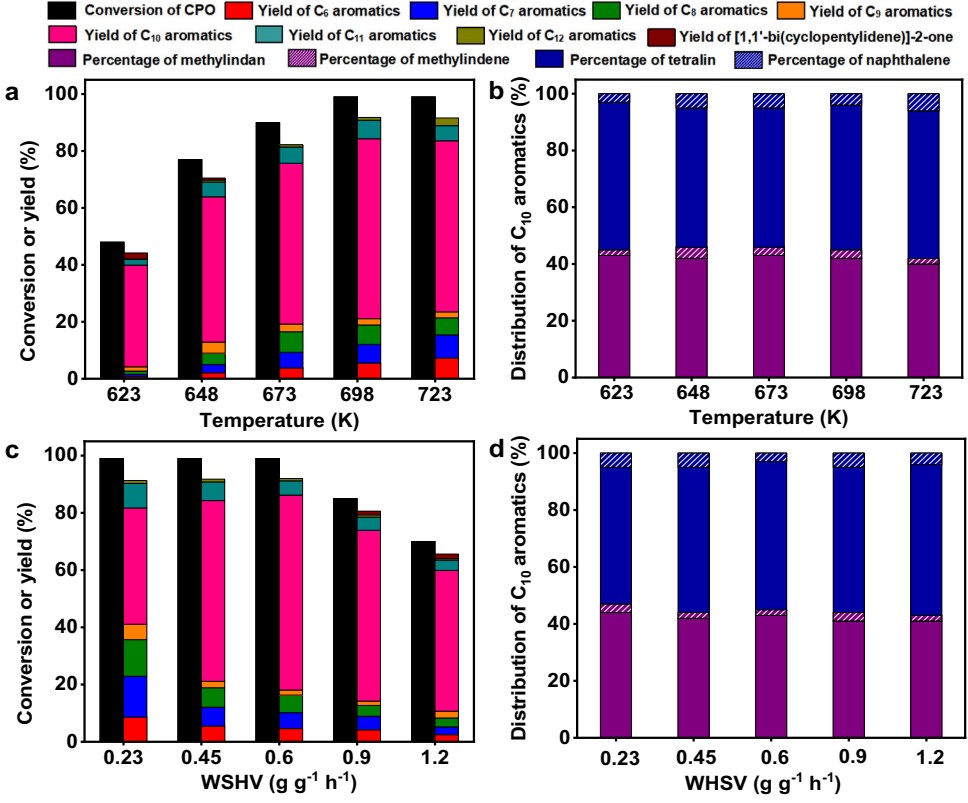

**Fig. 4 | Effects of reaction conditions on the catalytic conversion of CPO. a** CPO conversion and the carbon yields of products over the H-ZSM-5 (160) catalyst as the function of reaction temperature. **b** Distributions of $C_{10}$ aromatics (including methylindan, methylindene, tetralin and naphthalene) over the H-ZSM-5 (160) catalyst as the function of reaction temperature. **c** CPO conversion and the carbon yields of products over the H-ZSM-5 (160) catalyst as the function of WHSV. **d** Distributions of $C_{10}$ aromatics (including methylindan, methylindene, tetralin and naphthalene) over the H-ZSM-5 (160) catalyst as the functions of WHSV. Reaction conditions: 0.1 MPa $N_2$, WHSV = 0.45 g g$^{-1}$ h$^{-1}$, the initial $N_2$/CPO molar ratio = 36/1 **a**, **b**. 698 K, 0.1 MPa $N_2$, the initial $N_2$/CPO molar ratio = 36/1 **c**, **d**. CPO cyclopentanone, WHSV weight hour space velocity.

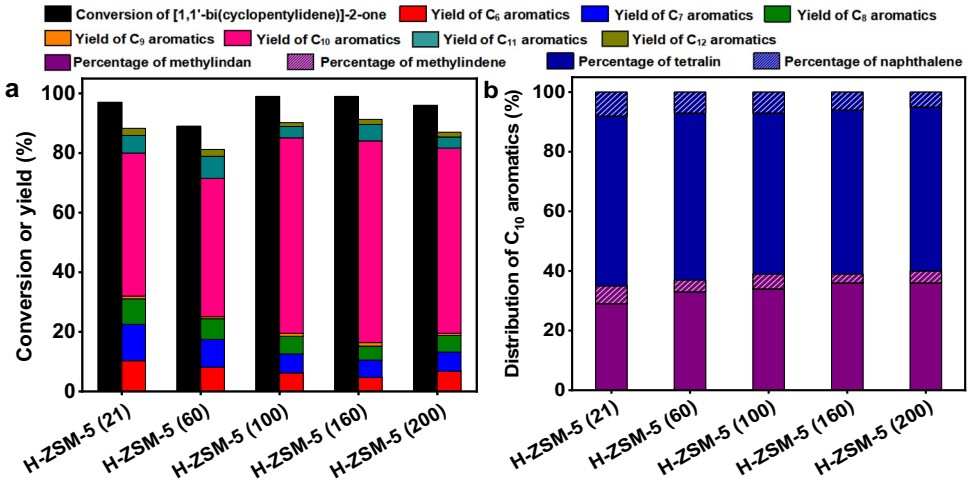

**Fig. 5 | Catalytic conversion of [1,1'-bi(cyclopentylidene)]−2-one over the H-ZSM-5 catalysts. a** [1,1'-bi(cyclopentylidene)]−2-one conversions and the carbon yields of products over the H-ZSM-5 catalysts with different SiO$_2$/Al$_2$O$_3$ molar ratios. **b** Distributions of C$_{10}$ aromatics (including methylindan, methylindene, tetralin and naphthalene) over the H-ZSM-5 catalysts with different SiO$_2$/Al$_2$O$_3$ molar ratios. Reaction conditions: 723 K, 0.1 MPa N$_2$, WHSV = 0.45 g g$^{-1}$ h$^{-1}$, the initial N$_2$/[1,1'-bi(cyclopentylidene)]−2-one molar ratio = 36/1. WHSV weight hour space velocity.

temperature (such as 623 K). In accordance with the Supplementary Figs. 32 and 33, some C$_{10}$ oxygenate (such as 3,4,5,6,7,8-hexahydronaphthalen-1(2*H*)-one (HHNO), 7a-methyl-5,6,7,7a-tetrahydro-1*H*-inden-2(4*H*)-one (MTHIO) and their isomers) were obtained from [1,1'-bi(cyclopentylidene)]−2-one rearrangement. These compounds have similar carbon chain structures as those of methylindan and tetralin. Therefore, they can be considered as the intermediates during the conversion of [1,1'-bi(cyclopentylidene)]−2-one (or CPO) to methylindan and tetralin over H-ZSM-5. On the basis of these results, a reaction mechanism for the generation of C$_6$-C$_{12}$ aromatics from the conversion of CPO over H-ZSM-5 was proposed in Fig. 6a. In the first step, [1,1'-bi(cyclopentylidene)]−2-one was generated by the acid-catalyzed self-aldol condensation of CPO. Subsequently, the [1,1'-bi(cyclopentylidene)]−2-one was rearranged to HHNO, MHHIO and their isomers. Finally, HHNO, MTHIO and their isomers were aromatized to methylindan and tetralin (by rearrangement, dehydration and hydrogen transfer reactions). Meanwhile, small amounts of methylindene and naphthalene were generated by the dehydrogenation of methylindan and tetralin. Likewise, some C-C cleavage (or cracking) reactions may also take place, which will lead to the generation of C$_6$-C$_9$ aromatics. The methyl fragments generated during the C-C cleavage (or cracking) reactions may react with the methylindan, tetralin, methylindene and naphthalene resulting in the generation of C$_{11}$ and C$_{12}$ aromatics.

Furthermore, we also checked the stability of the H-ZSM-5 (160) under the optimized reaction conditions. From the Fig. 7, it can be seen that the CPO conversion and the total carbon yield of methylindan and tetralin over the H-ZSM-5 (160) catalyst decreased with the increment of reaction time. After the in-situ regeneration by calcination in flowing air at 823 K for 4 h, the activity and selectivity of the H-ZSM-5 (160) catalyst restored to its initial level. Based on these results, we believe that the deactivation on the H-ZSM-5 (160) catalyst may be caused by the formation of coke during the reaction. In some recent literature[51,56,57], it has been suggested that nano H-ZSM-5 has smaller size and shorter pore length than those of conventional micrometer H-ZSM-5, which is favorable for the mass transfer and restrain the formation of coke. Therefore, we studied the conversion of CPO over a commercial nano-size H-ZSM-5 with the same SiO$_2$/Al$_2$O$_3$ molar ratio of 160 (denoted as nano H-ZSM-5 (160)). As we expected, the nano H-ZSM-5 (160) is very stable under the same conditions as we used for conventional micrometer H-ZSM-5 (160) catalyst. No evident decrease in activity was observed in the 24 h time-on-stream, which is advantageous in future application.

Based on the process flow diagram (Supplementary Fig. 34), the material balances and energy balances of the optimized reaction results (Supplementary Tables 5-7), two concise Sankey diagram were outlined for the production of methylindan and tetralin with xylose or hemicellulose, respectively (Supplementary Figs. 35 and 36). Meanwhile, we also compared the overall carbon yield and practical process of this work with what have been reported by our group[58] and other groups[59-63] for the synthesis of jet fuel range hydrocarbons with xylose or hemicellulose (Supplementary Tables 8 and 9). It was noticed that the overall carbon yield of jet fuel range hydrocarbons (53.6%) in this work was only second to the one (73.6%) achieved by Huber et al.[59] in their previous work about the synthesis of jet fuel range chain alkanes with hemicellulose. In addition to dehydration step, the overall carbon yield of jet fuel range aromatics can be increased to 60.3%. In future, more work will be carried out to improve the carbon yield and selectivity of CPO from the hydrogenolysis of hemicellulose or xylose. Meanwhile, we will also try to increase the carbon yield of jet fuel range aromatics from the reaction of CPO over zeolite catalyst.

### Synthesis of dimethyltetralin with cyclohexanone (CHO)

Analogous to CPO, CHO is also a cycloketone that can be produced from the selective hydrogenation of phenol from lignin. As an extension of this work, we also explored the applicability of H-ZSM-5 for the production of bicyclic aromatics with CHO. Based on Fig. 8 and Supplementary Figs. 37-39, the reaction of CHO over the H-ZSM-5 led to the selective production of C$_{12}$ bicyclic aromatics (including dimethyltetralin, dimethylnaphthalene, cyclohexylbenzene and their isomers) with dimethyltetralin as the main component. From the analysis of the product obtained over H-ZSM-5 at 613 K and a WHSV of 1.2 g g$^{-1}$ h$^{-1}$, [1,1'-bi(cyclohexylidene)]−2-one was identified as the intermediate between CHO and the C$_{12}$ aromatics (Supplementary Figs. 40 and 41). According to Fig. 6b, the dimethyltetralin may be generated by the cascade self-aldol condensation/rearrangement/aromatization reaction of CHO following a reaction mechanism similar to the one for the generation of tetralin with CPO. The dimethylnaphthalene was generated by the dehydrogenation of dimethyltetralin. The cyclohexylbenzene may be generated by the hydrogen transfer hydrodeoxygenation of the [1,1'-bi(cyclohexylidene)]−2-one from the self-aldol condensation product of CHO. Besides C$_{12}$ bicyclic aromatics, a series of C$_{10}$-C$_{11}$ bicyclic aromatics (including methylindan, methylindene, methyltetralin and methylnaphthalene) were also detected in the products. These C$_{10}$-C$_{11}$ bicyclic aromatics may be

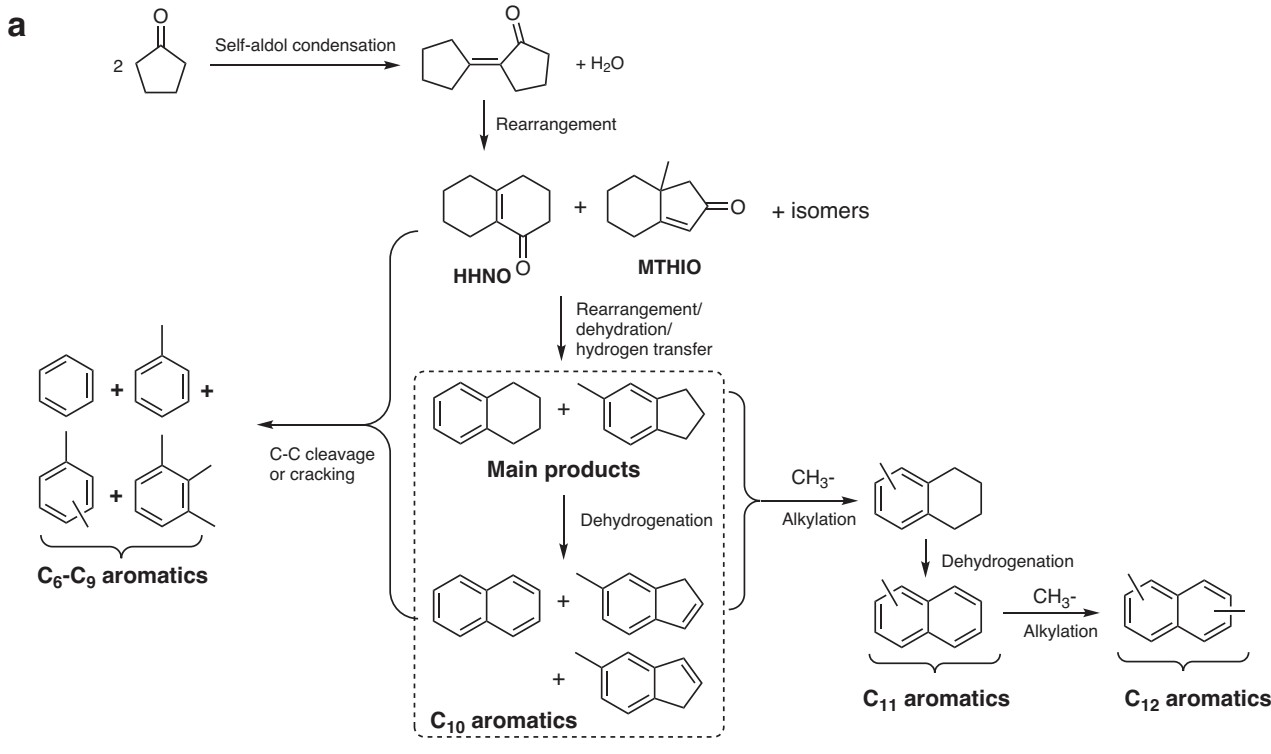

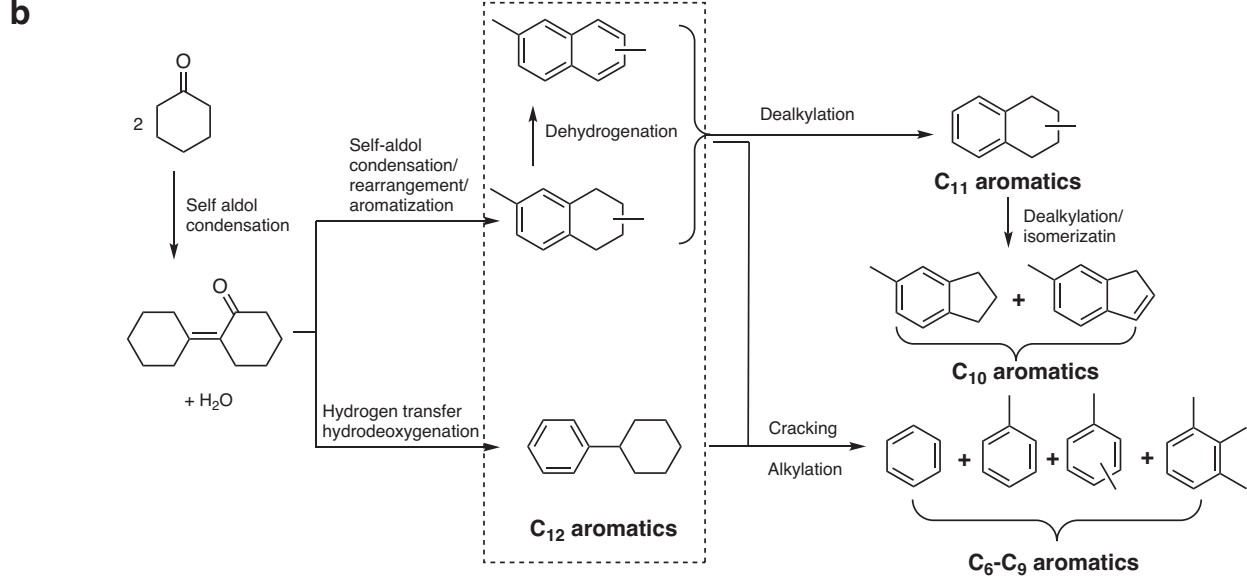

Fig. 6 | **Reaction pathways. a** Reaction pathway for the production of $C_6$-$C_{12}$ aromatics from CPO over the H-ZSM-5 catalysts. **b** Reaction pathway for the production of $C_6$-$C_{12}$ aromatics from CHO over the H-ZSM-5 catalysts. CPO: cyclopentanone. CHO: cyclohexanone. HHNO: 3,4,5,6,7,8-hexahydronaphthalen-1(2H)-one, MTHIO: 7a-methyl-5,6,7,7a-tetrahydro-1H-inden-2(4H)-one.

generated from the dealkylation (or dealkylation/isomerization) of the dimethyltetralin or dimethylnaphthalene. According to literature[10], the famous thermal stable jet fuel JP-900 is mixture of decalin, octahydroindene, bicyclohexane and their alkylation analogs. This work offers a brand-new method for the production of renewable thermal stable jet fuel with the cycloketones from lignocellulose. As we observed from the reaction of CPO, about 20% total carbon yield of $C_6$-$C_9$ aromatics was achieved from the conversion of CHO over the H-ZSM-5 (160) and nano H-ZSM-5 (160). As the potential applications, these $C_6$-$C_9$ aromatics can be either used as intermediates in the production of chemicals or hydrogenated to cyclohexane and alkylated

cyclohexane that have been utilized as additives to improve the thermal stability of jet fuel. Compared to conventional micrometer H-ZSM-5 (160), nano H-ZSM-5 (160) demonstrated slightly higher CHO conversion and selectivity to $C_{12}$ bicyclic aromatics. This may be explained by its small size that is favorable for mass transfer. Finally, we also checked the stability of the nano H-ZSM-5 (160). According to the result illustrated in Fig. 9, the nano H-ZSM-5 (160) is stable, no evident decrease in activity was observed in the 24 h time-on-stream. This result is consistent with what we observed in the conversion of CPO.

A two-step method was developed for the synthesis of methylindan and tetralin from xylose or hemicellulose. First, CPO was

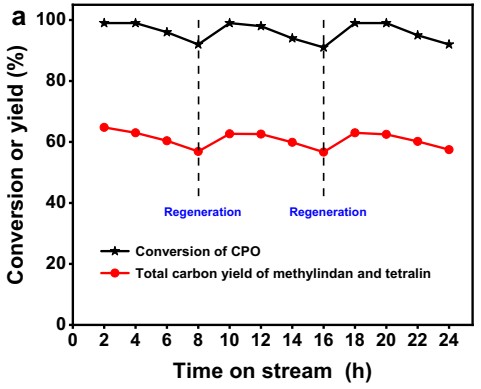
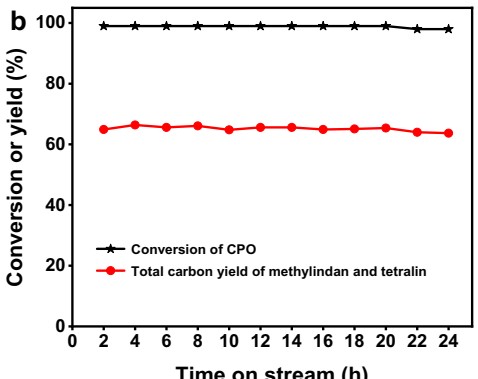

**Fig. 7 | Catalytic stability of the H-ZSM-5 catalysts in the conversion of CPO.** **a** CPO conversion and the total carbon yield of methylindan and tetralin over the H-ZSM-5 (160) catalyst as the function of time-on-stream. **b** CPO conversion and the total carbon yield of methylindan and tetralin over the nano H-ZSM-5 (160) catalyst as the function of time-on-stream. Reaction conditions: 698 K, 0.1 MPa $N_2$, WHSV = 0.6 g g$^{-1}$ h$^{-1}$, the initial $N_2$/CPO molar ratio = 36/1. CPO cyclopentanone, WHSV weight hour space velocity.

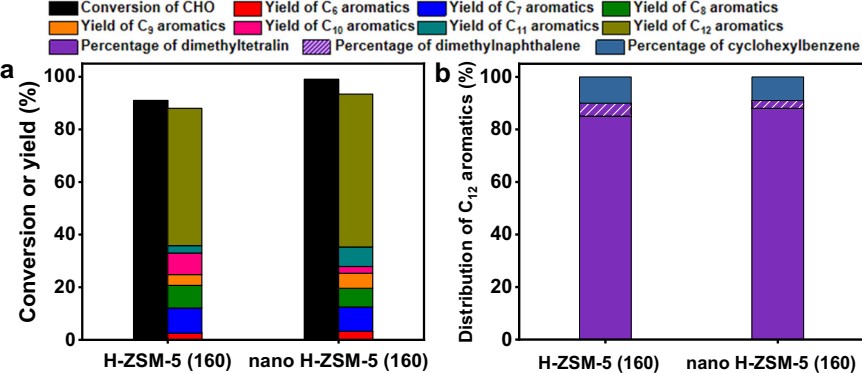

**Fig. 8 | Catalytic conversion of CHO over the zeolite catalysts. a** CHO conversion and the carbon yields of various products over the H-ZSM-5 (160) catalyst and the nano H-ZSM-5 (160) catalyst. **b** Distributions of $C_{12}$ aromatics over the H-ZSM-5 (160) catalyst and the nano H-ZSM-5 (160) catalyst. Reaction conditions: 663 K, 0.1 MPa $N_2$, WHSV = 0.6 g g$^{-1}$ h$^{-1}$, the initial $N_2$/CHO molar ratio = 36/1. CHO cyclohexanone, WHSV weight hour space velocity.

selectively prepared from the direct hydrogenolysis of xylose or hemicellulose using a non-noble bimetallic Cu-La/SBA-15 catalyst. The presences of NaCl aqueous solution and toluene promote the transformation of xylose to furfural by a cascade isomerization/dehydration reaction. Meanwhile, NaCl also restrained the hydrogenation of xylose and furfural to xylitol and tetrahydrofurfuryl alcohol. Therefore, high yield of CPO was achieved the Cu/SBA-15 catalyst. The doping of La increased the reuseablity of Cu/SBA-15 and made it a practical catalyst in real application. By a cascade self-aldol condensation/rearrangement/aromatization reaction, CPO can be selectively converted to methylindan and tetralin over H-ZSM-5 catalyst. Compared to conventional micrometer H-ZSM-5, nano H-ZSM-5 have higher stability which can be comprehended from the pointview of mass transfer. When we replaced CPO with CHO in the second step, the major products switched to $C_{12}$ bicyclic aromatics with dimethyltetralin as the main component. This paper gives an idea for the selective production of bicyclic aromatics with cheap and abundant lignocellulose.

## Methods
### Materials
The Cu(NO$_3$)$_2$, Ni(NO$_3$)$_2$, Co(NO$_3$)$_2$, La(NO$_3$)$_3$, Ce(NO$_3$)$_3$ and Y(NO$_3$)$_3$ were purchased from Aladdin Bio-Chem Technology Co.. Xylose, xylulose, and arabinose were purchased from Shanghai Aladdin Bio-Chem Technology Co. For real application, we also extracted the hemicellulose from poplar wood and corncob (two representatives of forest and agriculture resides) by the method reported by literature[64].

The experiments were carried out in a 50 mL stainless-steel batch reactor (Parr 5513 Stainless Steel, Parr Instrument). For each experiment, 1 g raw biomass, 10 mL methyltetrahydrofuran and 10 mL 0.1 mol L$^{-1}$ oxalic aqueous solution were used. Before the experiments, the reactor was purged with argon for five times to remove the air and heated to 428 K under stirring (at a rotation speed of 400 rpm). After being stirred at 428 K for 2.5 h and quenched to room temperature with cool water, the mixture was taken out from the reactor. The cellulose (obtained as solid residue) was separated from the reaction system by filtration, the lignin was solved in the methyltetrahydrofuran phase, the hemicellulose was obtained in the aqueous phase. Subsequently, the water in the aqueous phase was removed by vacuum distillation at 343 K and the solid products was repeatedly washed by ethanol for several times to remove the oxalic acid. After being dried at 353 K overnight, the solid products were dissolved in 10 mL water and analyzed by HPLC before they were used for the activity test. After being used, most of the methyltetrahydrofuran existed in organic phase, which can be rationalized by its low solubility in water. Therefore, it could be easily separated with lignin by the vacuum distillation and reused for the extraction of hemicellulose. Likewise, the water and small part of methyltetrahydrofuran in aqueous phase product could be separated with the extracted hemicellulose by vacuum distillation and reused for the separation of hemicellulose.

The H-ZSM-5 zeolites with different SiO$_2$/Al$_2$O$_3$ molar ratios (denoted as H-ZSM-5 (X), and X is the SiO$_2$/Al$_2$O$_3$ molar ratio of H-ZSM-5), H-Y, H-β, H-USY, and SAPO-34 zeolites were purchased from NanKai

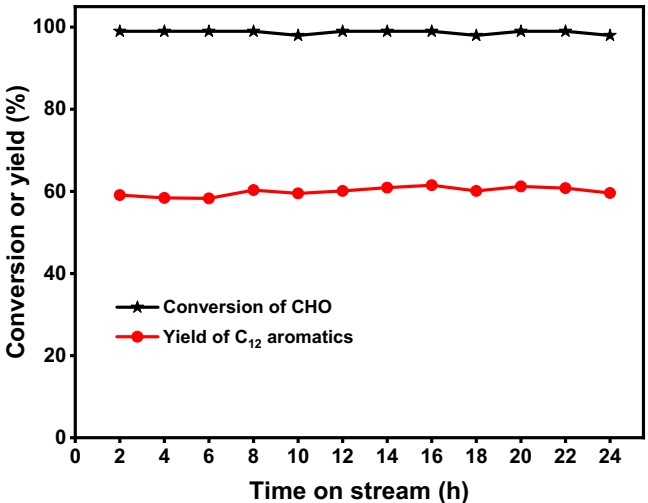

**Fig. 9 | Catalytic stability of the nano H-ZSM-5 (160) catalyst in the conversion of CHO.** Reaction conditions: 663 K, 0.1 MPa $N_2$, WHSV = 0.6 g g$^{-1}$ h$^{-1}$, the initial $N_2$/CHO molar ratio = 36/1. CHO cyclohexanone, WHSV weight hour space velocity.

University Catalyst Co., Ltd. According to the information from the supplier, the $SiO_2/Al_2O_3$ molar ratios of H-Y, H-β, and H-USY zeolites were 5.4, 25, and 11, respectively. Prior to the activity tests, the zeolite catalysts were calcined at 823 K for 4 h in muffle furnace to remove the water or impurity in them. For comparison, the nano H-ZSM-5 catalyst with a $SiO_2/Al_2O_3$ molar ratios of 160 (denoted as nano H-ZSM-5 (160) catalyst in the manuscript) was purchased from Nanjing XFNANO Materials Tech Co., Ltd. According to the information from the supplier, the average size of the nano H-ZSM-5 (160) catalyst was 100 nm.

## Preparation of catalysts

The Cu/SBA-15, Ni/SBA-15 and Co/SBA-15 catalysts with theoretical metal content of 30 wt% were synthesized by the ammonia evaporation method. Typically, nitrate salts were dissolved in 100 mL of deionized water and the aqueous ammonia was used to adjust the initial pH to 11–12. Subsequently, a commercial SBA-15 support was added to the solution. The mixture was then heated to 368 K for the evaporation of ammonia. When the pH of the suspension decreased to 6–7, the precipitate was filtered and dried overnight. Finally, the precursors were calcined at 673 K under Air for 4 h. Before the activity tests, the catalysts were reduced at 673 K in 20% $H_2$/Ar flow for 4 h. The Cu-La/SBA-15, Cu-Ce/SBA-15 and Cu-Y/SBA-15 catalysts with theoretical rare-earth metal contents of 3 wt% were prepared the incipient wetness impregnation of the Cu/SBA-15 with the aqueous solution of La(NO$_3$)$_3$, Ce(NO$_3$)$_3$ and Y(NO$_3$)$_3$, respectively. The products were dried overnight and calcined at 673 K in air flow for 4 h. Before the activity tests, the catalysts were reduced at 673 K in 20% $H_2$/Ar flow for 4 h. The NaCl-treated Cu-La/SBA-15 was prepared by stirring 1 g of the Cu-La/SBA-15 catalyst in 30 mL of 3 wt% NaCl aqueous solution at 433 K for 4 h in a batch reactor. Before the treatment, the reactor was purged by $N_2$ three time. After cooling down the reactor to room temperature, the NaCl treated Cu-La/SBA-15 catalyst was separated from reaction system by filtration, washed with deionized water, and dried overnight at 333 K.

## Activity test

The hydrogenolysis of xylose, xylulose, arabinose, xylan, and the extracted hemicelluloses was carried out in an organic-aqueous biphasic reaction system with a 50 mL stainless-steel batch reactor. Typically, the substrate, 10 mL salt aqueous solution, 10 mL organic solvents, and catalysts were loaded into the reactor. After being purged with $H_2$ 5 times, the reactor was charged with $H_2$ to 3 MPa (at

room temperature). Subsequently, the reactor was heated to the set temperature within 40 min at a stirring rate of 500 rpm and hold at that temperature for 4 h. After the activity test, the reactor was quenched to room temperature by cool water. The biphasic product was taken out from the reactor and separated with catalyst (and the unreacted xylan) by filtration. The organic phase was analyzed by a gas chromatography (GC). The aqueous phase was analyzed by a high performance liquid chromatography (HPLC). The conversions of substrates were calculated by their consumption during the reaction. The carbon yield of specific product was calculated using the equation: carbon yield (%) = (carbon in the specific product)/(carbon in the substrate use in the test) × 100%. Based on our analysis, the cyclopentanone (CPO), cyclopentanol (CPL), 2-methylfuran (MF), furfural (FF), furfuryl alcohol (FA), and tetrahydrofurfuryl alcohol (THFA) generated in the hydrogenolysis reaction mainly existed in organic phase. The concentrations of these compounds in aqueous phase products were so low that they could not be detected by HPLC or GC. This phenomenon can be comprehended by the salt-out effect of NaCl that has been reported in the previous work of Dumesic et al.[48].

The catalytic conversion of CPO (or CHO) was conducted in a tubular fixed-bed reactor made of 316 L stainless steel. To maintain a constant bed length, the catalysts were physically diluted by inert quartz granules (40–70 meshes). Dense quartz granules (20–40 mesh) were used to pack the rest of the reactor and quartz wool plugs were used to seal the both ends of the reactor. Before the activity tests, the catalysts were heated in $N_2$ flow at reaction temperature for 0.5 h, then the liquid feedstock was co-fed with the $N_2$ flow from the top of the reactor to the catalyst bed using a HPLC pump. The gas phase products were analyzed by an online Agilent 7890B GC, and the liquid products were taken out from the separator after 2 h, diluted with ethanol and immediately analyzed by an Agilent 7890 A GC using 1,4-dioxane as the internal standard. Conversion of CPO (or CHO) (%) = (1−mole of unreacted CPO (or CHO)/mole of CPO (or CHO) that was fed into the reactor) × 100%; Carbon yield of a specific product (%) = (mole of carbon in a specific product/mole of carbon in the converted CPO (or CHO)) × 100%. The percentages of specific products in $C_{10}$ (or $C_{12}$) aromatics were calculated by following equation: percentage of a specific product in $C_{10}$ (or $C_{12}$) aromatics (%) = (mole of carbon in a specific $C_{10}$ (or $C_{12}$) aromatic product/mole of carbon in all of the $C_{10}$ (or $C_{12}$) aromatics obtained from the reaction) × 100%.

## Characterization of catalysts

Thermogravimetric-Differential Thermal Analysis-Mass spectrometer (TG-DTA-MS) tests of the used bimetallic catalysts were carried out by the TA Instrument SDT Q 600 connected with an InProgress Instruments GAM 200 mass spectrometer (MS). The elemental distribution of the catalysts was analyzed by scanning transmission electron microscopy (STEM) equipped with an energy dispersive X-ray spectroscopy (EDX) system. For each test, 8 mg catalyst was used. The experiments were conducted in flowing air (at a rate of 100 mL min$^{-1}$) from 303 K to 873 K at a heating rate of 10 K min$^{-1}$. Nitrogen physisorption of the zeolite catalysts were carried out by a Micromeritics ASAP 2010 apparatus. $NH_3$-chemisorption and the temperature-programmed desorption of $NH_3$ ($NH_3$-TPD) of zeolite catalysts were carried out on a Micromeritics Autochem 2910 chemisorption apparatus. Typically, 0.1 g of zeolite catalyst was used for the tests. Before the test, the zeolite catalyst was pretreated in a quartz reactor at 573 K under He flow for 0.5 h and cooled in He flow to 373 K. After the baseline was stabilized, the $NH_3$ was injected into the reactor by manual pulse until the zeolite catalyst was adsorbed to saturation. The pulse signal was detected by a thermal conductivity detector (TCD). The amounts of acid sites on the zeolite catalysts were calculated by the consumption of $NH_3$. Subsequently, the zeolite catalyst was purged in He for 10 min to remove the physically adsorbed $NH_3$. $NH_3$-TPD was carried out in He flow at a heating rate of 10 K min$^{-1}$ from 373 K to

1073 K. The desorbed NH$_3$ was monitored by OminiStar mass spectrometer. The Fourier Transform infrared (FT-IR) spectra of H-ZSM-5 catalysts were recorded (at a resolution of 4 cm$^{-1}$) on a Bruker spectrometer equipped with liquid-nitrogen cooled MCT detector using pyridine as the probing molecule. Prior to pyridine adsorption, the solid acid catalysts were evacuated at 723 K for 0.5 h, then cooled down to room temperature. At this stage, the spectra were collected as the background references. Subsequently, pyridine was introduced for the adsorption for 5 min at room temperature followed by evacuation for 30 min. Finally, the temperature was elevated to 423 K (or 623 K) and the system was evacuated for 30 min. All the spectra were collected at room temperature and extracted with the background reference.

## Data availability
The data that support the findings of this study are available within the paper and its Supplementary Information and all data are available from the corresponding authors on request.

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

## Acknowledgements

This work was supported by the National Key R&D Program of China (no. 2022YFA1504902 received by N.L.), National Natural Science Foundation of China (no. 22178335 received by N.L.; 21721004 received by T.Z.), Joint Fund of the Yulin University and the Dalian National Laboratory for Clean Energy (Grant: YLU-DNL Fund 2021020 received by N.L.).

## Author contributions

Z.Z. and Z.Y. contributed equally to this work. Z.Y. performed the catalyst preparation and activity tests for the selective hydrogenolysis of xylose and extracted hemicellulose to cyclopentone (CPO) over Cu-based catalysts. Z.Z. conducted the activity tests and reaction mechanism exploration for the conversion of CPO to methylindan and tetralin. Meanwhile, Z.Z. also conducted the activity tests and reaction mechanism exploration for the conversion of cyclohexanone (CHO) to $C_{12}$ bicyclic aromatics (including dimethyltetralin, dimethylnaphthalene, cyclohexylbenzene, and biphenyl) with dimethyltetralin as the main component. W.G. prepared the Cu-based bimetallic catalysts. Y.L., Y.Y., Y.H., and X.L. analyzed the products. G.L., A.W. and Y.C. analyzed the data. N.L. and T.Z. conceived the overall direction of the project. Z.Y., Z.Z., N.L. and T.Z. co-wrote the paper. All the authors discussed the results and provided input for the manuscript.

## Competing interests

The authors declare no competing interests.
