## [Peer Review File · Nature Communications]

Selective production of methylindan and tetralin with xylose or hemicelluloseEditorial note: Parts of this Peer Review File have been redacted as indicated to remove third-party material where no permission to publish could be obtained.

REVIEWER COMMENTS

Reviewer #2 (Remarks to the Author):

This manuscript reports conversion of xylose to methylindan and tetralin via cyclopentanone for biomass utilization. This manuscript is highly original and has academic value. There are some concerns to be addressed before this article is accepted for publication in Nature Communications.

In the case of xylose conversion, the selectivity was drastically changed by the concentration of NaCl.

NaCl is effective in extracting substances that are soluble in organic solvents. Therefore, xylose exists in water regardless of the presence of NaCl. So, the effect of NaCl is strange. It seems that NaCl is effective not only for extraction but also for catalysis. The authors should discuss that NaCl changed selectivity of xylose conversion.

Reviewer #3 (Remarks to the Author):

This manuscript reports the synthesis of bicyclic aromatics from CPO, itself obtained from C5-sugar feedstocks. The first part (production of CPO) appears more as an optimization of previous works (replacement of noble by non-noble metals) rather than a real breakthrough in the field. The second part, dealing with the conversion of CPO to aromatics over zeolites, is in my view more original. Although the work is definitely of interest, similar reaction pathways have been reported in the current literature, for instance by Pradeep K. Agrawal and Christopher W. Jones (*J. Phys. Chem. C* 2009, 113, 16702–16710). Furthermore, compounds were only characterized by means of MS and GC. NMR, or at least HRMS, should be used to confirm the structure of chemicals.

In its present form, I do regret to not recommend acceptance of this work for publication in

Nat. Com. I recommend submission to a journal more specialized on catalysis.

Reviewer #4 (Remarks to the Author):

This paper demonstrates a two step catalytic route for production of methylindan and tetralin from xylose. Xylose is first catalytically converted into cyclopentanone (CPO) with a Cu-La/SBA-15 catalyst. The CPO is then converted into methylindan and tetralin over a ZSM5 catalyst. The authors have lots of experiments for these two reactions in a batch and continuous flow reactor. The authors claim that the methylindan and tetralin could be used as jet fuel. My chief concerns with this paper is that I do not think this route would ever be viable to produce jet fuel because 1) a low overall yields is obtained (0.36 overall carbon yield) and 2) they are doing the first step in a NaCl and water solution and it will be expensive to purify the water and do waste water treatment. Nevertheless the authors have reported some unique chemistry. As the authors state (and have previously published on) CPO is a very valuable intermediate to produce flavors and fragrances. A number of other articles have been published in the literature to convert xylose into jet or diesel fuel including by this same group. The authors needs to do a rigorous comparison (in terms of overall yields and practical processes) for this new approach and the other approaches in the literature in any article they publish. They have neglected a comparison with any other approaches in the literature. I would also expect to see a Sankey diagram or a report of the overall yield of products. They need to discuss what happens to the water and NaCl that is in this process. I am on the fence about this paper but willing to consider a revision.

Here are a couple of other minor comments about this article:

1. The authors state that they have done recyclability tests of the catalysts. However all the recycling tests are done at high conversion where deactivation can be hidden. They should either characterize the catalyst after reaction to see if leaching of the Cu or change of Cu particle size occurred or they should do the stability tests at lower conversion where you can actually observe deactivation.
2. The cartoon in Scheme 1 need to be improved and made more professional. My opinion (which could be wrong) are that the cartoons are not appropriate for a technical journal. I have no idea why a pill is running. The authors need to show what happens to the large

amount of H₂O and NaCl that are used in this product. They also need to show any other byproducts.

3. What catalysts was used in Figure 1a? I assume it is Cu/SAB-15 as all the other captions state. The writing in the paper needs to be slightly improved to bring it up to journal standards. In Figures 3-5 it is very hard for me to tell the methylindene and naphthalene from the captions in the key. I suggest making those captions easier to read.

4. The authors conclude with this statement, "A two-step method was developed for the manufacture of methylindane and tetralin from xylose or hemicellulose." I think they are far from manufacturing these products in this route. Hemicellulose is a lot more complicated than just xylose. There are really no commercially available xylose streams on the market today. The authors need to show a process flow diagram with material and energy balances to state they could manufacture these products. The authors in this paper focus on the basic process chemistry which is an important first step but it is not correct to claim they have an actual process at this early stage.

Point-by-point response to the referees' comments

Response to the reviewers' comments

First of all, we appreciate all of the three reviewers for their in-depth comments and constructive suggestions. Frankly, they really helped us to improve the quality of this manuscript. We have made point-to-point responses and revised the manuscript accordingly with all changes being highlighted in the revised manuscript.

Reviewer # 2 (Remarks to the Author):

This manuscript reports conversion of xylose to methylindan and tetralin via cyclopentanone for biomass utilization. This manuscript is highly original and has academic value. There are some concerns to be addressed before this article is accepted for publication in Nature Communications.

Response: Thanks a lot for your good comments to our manuscript. Following your constructive suggestion, we made some revisions to our manuscript, we hope that you will be satisfied with it.

Comment 1. *In the case of xylose conversion, the selectivity was drastically changed by the concentration of NaCl. NaCl is effective in extracting substances that are soluble in organic solvents. Therefore, xylose exists in water regardless of the presence of NaCl. So, the effect of NaCl is strange. It seems that NaCl is effective not only for extraction but also for catalysis. The authors should discuss that NaCl changed selectively of xylose conversion.*

Response: To figure out the reasons for the significant promotion effect of NaCl, we did some additional experiments. Firstly, we studied the conversion of xylose in the absence of metal catalyst under the investigated reaction conditions. From the GC and HPLC chromatograms of the products that were obtained in a toluene/NaCl aqueous solution biphasic reaction system at a short reaction time of 35 min (see Figures R1 and R2), we can clearly see the peaks of xylulose and furfural. In contrast, we didn't notice these peaks in the GC and HPLC chromatograms of the products that were obtained in a toluene/water biphasic reaction system. Based on these results, we believe that the presence of NaCl accelerated the isomerization/dehydration of xylose to furfural. This may be one reason for the promotion effect of NaCl.

Figure R1. GC and HPLC chromatograms of the products obtained from the reaction of xylose in a toluene/NaCl aqueous solution biphasic reaction system. Reaction conditions: $T = 433$ K, $P_{N_2} = 0.5$ MPa, 500 rpm, 35 min; 10 mL 5 wt.% NaCl aqueous solutions, 10 mL toluene and 0.3 g xylose were used for the test.

Figure R2. GC and HPLC chromatograms of the products obtained from the reaction of xylose in a toluene/H₂O biphasic reaction system. Reaction conditions: $T = 433$ K, $P_{N_2} = 0.5$ MPa, 500 rpm, 35 min; 10 mL dionized H₂O, 10 mL toluene and 0.3 g xylose were used for the test.

Secondly, we compared the catalytic performances of the fresh Cu-La/SBA-15 and the NaCl treated Cu-La/SBA-15 for the hydrogenation of furfural in toluene. Based on the results illustrated in Figures

R3a and R3b, furfural was completely hydrogenated to tetrahydrofurfuryl alcohol over the fresh Cu-La/SBA-15 catalyst, while a mixture of furfuryl alcohol and tetrahydrofurfuryl alcohol was obtained over the NaCl treated Cu-La/SBA-15 catalyst. When the hydrogenation of furfural was carried out in a toluene/H₂O biphasic reaction system (see Figure R3c), CPO was obtained as the main product over the NaCl treated Cu-La/SBA-15 catalysts. Based on above results, we can draw two conclusions: 1) the presence of Cl⁻ ion in the reaction system restrains the hydrogenation of furan ring over Cu-La/SBA-15, which may be another reason for the promotion effect of NaCl; 2) the presence of water is necessary for the generation of CPO from the hydrogenation of furfural.

Figure R3. GC chromatograms of the products obtained from the hydrogenation of furfural over the (a) fresh and (b, c) NaCl-treated Cu-La/SBA-15 catalysts. Reaction conditions: (a) $T = 433$ K, $P_{H_2} = 3$ MPa, 500 rpm, 4 h; 10 mL toluene, 0.3 g furfural and 30 mg fresh Cu-La/SBA-15 catalyst were used for the test. (b) $T = 433$ K, $P_{H_2} = 3$ MPa, 500 rpm, 4 h; 10 mL toluene, 0.3 g furfural and 30 mg NaCl treated Cu-La/SBA-15 catalyst were used for the test. (c) $T = 433$ K, $P_{H_2} = 3$ MPa, 500 rpm, 4 h; 10 mL deionized water, 10 mL toluene, 0.3 g furfural and 30 mg NaCl treated Cu-La/SBA-15 catalyst were used for the test.

Figure R4. HPLC chromatograms of the products from the hydrogenation of xylose over the fresh and NaCl-treated Cu-La/SBA-15 catalysts. Reaction conditions: $T = 378$ K, $P_{H_2} = 3$ MPa, 500 rpm, 4 h; 10 mL deionized water, 0.3 g xylose and 30 mg fresh or NaCl treated Cu-La/SBA-15 catalyst were used for the test.

Thirdly, we also compared the activities of the fresh and NaCl treated Cu-La/SBA-15 catalysts for the hydrogenation of xylose. Based on the results illustrated in Figure R4, the presence of Cl⁻ ion in the reaction system restrained the hydrogenation of xylose to xylitol, which may be the third reason for the promotion effect of NaCl.

To understand the poisoning effect of NaCl on the hydrogenation of furfuryl alcohol to tetrahydrofurfuryl alcohol and the hydrogenation of xylose to xylitol, we characterized the fresh, NaCl treated and used Cu-La/SBA-15 catalysts. According to TEM-EDX elemental mappings results (Figure R5), the metal particles on the surface of Cu-La/SBA-15 catalyst were partially covered by Cl species during the NaCl treatment or usage in the toluene/NaCl aqueous solution biphasic reaction system, which may restrain the planar adsorption and hydrogenation of furan ring. From the XPS results illustrated in Figure R6, it is also noticed that the Cu 2p band energy of the Cu-La/SBA-15 catalyst shifted to higher values after it was treated by the NaCl aqueous solution or used for the hydrogenolysis of xylose in a toluene/NaCl aqueous solution biphasic system which can be rationalized by the electron transfer from Cu species to the Cl species adsorbed on metal particles. Following your suggestion, we have added these results as Supplementary Figs. 6, 7 and 11-14 in the updated supporting information. Some comments were also given at the pages pages 4 and 6 of the revised manuscript.

Figure R5. TEM-EDX elemental mappings results of the NaCl treated and the used Cu-La/SBA-15 catalysts.

Figure R6. Cu 2p XPS spectra of the fresh, NaCl treated and used Cu-La/SBA-15 catalysts.

Reviewer # 3 (Remarks to the Author):

This manuscript reports the synthesis of bicyclic aromatics from CPO, itself obtained from C₅-sugar feedstocks. The first part (production of CPO) appears more as an optimization of previous works (replacement of noble by non-noble metals) rather than a real breakthrough in the field. The second part, dealing with the conversion of CPO to aromatics over zeolites, is in my view more original. Although the work is definitely of interest, similar reaction pathways have been reported in the current literature, for instance by Pradeep K. Agrawal and Christopher W. Jones (J. Phys. Chem. C 2009, 113, 16702 - 16710). Furthermore, compounds were only characterized by means of MS and GC. NMR, or at least HRMS, should be used to confirm the structure of chemicals.

In its present form, I do regret to not recommend acceptance of this work for publication in Nat. Com. I recommend submission to a journal more specialized on catalysis.

Response: Thanks for your kind reminding. We appreciate your comments which are very helpful in improving the quality of our manuscript. For the originality of synthesis of bicyclic aromatics with CPO, we have perused the reference you mentioned (Pradeep K. Agrawal and Christopher W. Jones, J. Phys. Chem. C 2009, 113, 16702-16710) and found that that work is totally different from ours.

Different kind of research: The literature you mentioned is a qualitative research based on the ¹³C and ²⁹Si MAS NMR spectra. Due to the limitation of research method, this work only pointed out the possibility to produce bicyclic aromatics by the reaction of cyclopentanone (CPO) over H-Y zeolite. Neither CPO conversions nor the yields of different products were given in that work. In contrast, our work was a quantitative research carried out in a fix-bed reactor over H-ZSM-5 catalyst. Based on the analysis of products, the CPO conversion and the carbon yields of different products were given, which can help the readers have a clearer idea about this reaction. The stability of H-ZSM-5 catalyst was investigated as well. Following the reviewer's suggestion, we separated the bicyclic aromatics from the product under the help of Prof. Xinmiao Liang's group in our institute. Based on the NMR spectra (see Figures R7 and R8), the major C₁₀ aromatics products from the reaction of CPO over the H-ZSM-5 (160) catalyst was 5-methyl-2,3-dihydro-1*H*-indene and tetralin. We have added their NMR spectra as Supplementary Figs. 20 and 21 in the updated supporting information. Analogously, we also separated the major C₁₂ bicyclic aromatics obtained from the reaction of CHO over the H-ZSM-5 (160) catalyst. Based on the NMR spectra (see Figure R9), the major C₁₂ aromatics products obtained in this

work was dimethyltetralin. We have added its NMR spectra as Supplementary Fig. 37 in the updated supporting information.

Figure R7 ^1H NMR (a) and ^{13}C NMR (b) spectra of the 5-methyl-2,3-dihydro-1H-indene (abbreviated as methylindan) obtained from the reaction of CPO over the H-ZSM-5 (160) catalyst.

Figure R8 ^1H NMR (a) and ^{13}C NMR (b) spectra of the tetralin obtained from the reaction of CPO over the H-ZSM-5 (160) catalyst.

Figure R9 ^1H NMR (a) and ^{13}C NMR (b) spectra of the dimethyltetralin obtained from the reaction of CHO over the H-ZSM-5 (160) catalyst.

Different reaction mechanisms: In the literature you mentioned, it was suggested that the C₁₀ aromatics were generated by the further reaction of CPO trimer (generated from deep condensation of [1,1'-bi(cyclopentylidene)]-2-one and CPO) following a reaction mechanism illustrated in Figure R10. In our manuscript, it was suggested that the C₁₀ aromatics were generated by the further reaction of [1,1'-bi(cyclopentylidene)]-2-one following a reaction mechanism illustrated in Figure R11. To verify our reaction mechanism, we conducted some additional experiments using CPO, [1,1'-bi(cyclopentylidene)]-2-one or their mixture (at the initial molar ratio of 1:1) as the feedstocks, respectively. If the reaction mechanism proposed in literature is correct, higher yield of C₁₀ aromatics will be generated when we used the [1,1'-bi(cyclopentylidene)]-2-one and CPO as the feedstock (because when we used CPO as the feedstock, CPO need to be converted to [1,1'-bi(cyclopentylidene)]-2-one by self aldol condensation then react with CPO, while [1,1'-bi(cyclopentylidene)]-2-one need to be converted to CPO by retro-aldol condensation then react with [1,1'-bi(cyclopentylidene)]-2-one). According to Figure R12, higher C₁₀ aromatics carbon yield (or selectivity) was achieved when we used [1,1'-bi(cyclopentylidene)]-2-one as the feedstock under the investigated conditions.

Figure R10. Reaction mechanism proposed in literature.

Figure R11. Reaction mechanism proposed in our manuscript.

Figure R12. Conversions of CPO or [1,1'-bi(cyclopentylidene)]-2-one, the carbon yields of products and the distributions of C₁₀ aromatics (including methylindan, methylindene, tetralin and naphthalene) over the H-ZSM-5 (160) catalyst. Reaction conditions: WHSV = 0.45 g g⁻¹ h⁻¹, T = 723 K, P_{N₂} = 0.1 MPa, the initial N₂/substrates molar ratio = 36/1.

Furthermore, we also compared the reactivity of CPO, [1,1'-bi(cyclopentylidene)]-2-one and the mixture of [1,1'-bi(cyclopentylidene)]-2-one and CPO (the initial molar ratio = 1:1) at lower reaction temperature. Based on Figure R13, [1,1'-bi(cyclopentylidene)]-2-one has higher reactivity than those of CPO and the mixture of [1,1'-bi(cyclopentylidene)]-2-one and CPO. Based on above results, we believe that our reaction mechanism is correct.

Figure R13. Conversions of CPO or [1,1'-bi(cyclopentylidene)]-2-one, the carbon yields of products and the distributions of C₁₀ aromatics (including methylindan, methylindene, tetralin and naphthalene) over the H-ZSM-5 (160) catalyst. Reaction conditions: WHSV = 0.45 g g⁻¹ h⁻¹, T = 673 K, P_{N₂} = 0.1 MPa, the initial N₂/substrate molar ratio = 36/1.

Reviewer # 4 (Remarks to the Author):

This paper demonstrates a two step catalytic route for production of methylindan and tetralin from xylose. Xylose is first catalytically converted into cyclopentanone (CPO) with a Cu-La/SBA-15 catalyst. The CPO is then converted into methylindan and tetralin over a H-ZSM-5 catalyst. The authors have lots of experiments for these two reactions in a batch and continuous flow reactor.

Response: Thanks for your good comments to our manuscript. Following your suggestions, we did some additional experiments and made some modification to our manuscript. We hope that you can be satisfied with it.

The authors claim that the methylindan and tetralin could be used as jet fuel. My chief concerns with this paper is that I do not think this route would ever be viable to produce jet fuel because 1) a low overall yields is obtained (0.36 overall carbon yield) and 2) they are doing the first step in a NaCl and water solution and it will be expensive to purify the water and do waste water treatment.

Response: Thanks for your reminding. Jet fuel is one of the potential applications of methylindan and tetralin. We think the best application of methylindan and tetralin should be chemicals or fuel additives

that are more valuable than jet fuel. Furthermore, we want to clarify that 36% is just the overall carbon yield of methylindan and tetralin from xylose. Besides methylindan and tetralin, the C₆-C₉ and C₁₁-C₁₂ aromatics obtained in the second step can also be considered as jet fuel range aromatics. If we take into account of these jet fuel range aromatics, the actual carbon yield of jet fuel rang aromatics can be higher (53.6%). From Figure 2 and Table 1, we can see that cyclopentanol (CPL) was also obtained as by-product during the hydrogenolysis of xylose to cyclopentanone (CPO). In real application, we can further increase the CPO yield by the dehydrogenation. To verify this hypothesis, we did an additional experiment using a commercial Pd/C catalyst. After the further dehydrogenation of the toluene phase product from the hydrogenolysis of xylose at 463 K for 4 h (see Figure R14), the CPO carbon yield was improved from 58.5% to 65.8%. By this way, the overall carbon yield of jet fuel rang aromatics was improved from 53.6% to 60.3%.

Figure R14. Xylose conversion and the carbon yields of different products from (a) hydrogenolysis of xylose over the Cu-La/SBA15 catalyst or (b) hydrogenolysis of xylose over the Cu-La/SBA15 catalyst and dehydrogenation of toluene phase product from the hydrogenolysis step over Pd/C catalyst. Reaction conditions: (a) $T = 433$ K, $P_{H_2} = 3$ MPa, 500 rpm, 4 h; 10 mL 3 wt.% NaCl aqueous solution, 10 mL toluene, 0.3 g xylose and 30 mg catalyst were used for the test. (b) First step: $T = 433$ K, $P_{H_2} = 3$ MPa, 500 rpm, 4 h; 10 mL 3 wt.% NaCl aqueous solution, 10 mL toluene, 0.3 g xylose and 30 mg catalyst were used for the test. Second step: $T = 463$ K, $P_{N_2} = 1$ MPa, 500 rpm, 4 h, 10 mL toluene phase product from the first step and 20 mg Pd/C were used for the test.

Due to the salt-out effect, most of the organic products (except for xylitol and xylose) were extracted into toluene phase. As we know, NaCl, xylitol and xylose are edible. Therefore, we needn't to worry too much about the wastewater. Following your suggestion, we also checked the reusability of NaCl aqueous solution (see Figure R15). As we expected, the NaCl aqueous solution can be repeatedly used without any treatment, which is advantageous in real application. Following your suggestion, we have added these results as Supplementary Figs. 15 and 16 in the updated supporting information.

Figure R15. Xylose conversion and the carbon yields of different products from hydrogenolysis of xylose over the Cu-La/SBA-15 catalyst as the function of recycle time of NaCl aqueous solution. Reaction conditions: $T = 433$ K, $P_{H_2} = 3$ MPa, 500 rpm, 4 h; 10 mL toluene, 10 mL 3 wt.% NaCl aqueous solution, 0.3 g xylose and 30 mg Cu-La/SBA-15 catalysts were used for the test.

Nevertheless the authors have reported some unique chemistry. As the authors state (and have previously published on) CPO is a very valuable intermediate to produce flavors and fragrances. A number of other articles have been published in the literature to convert xylose into jet or diesel fuel including by this same group. The authors needs to do a rigorous comparison (in terms of overall yields and practical processes) for this new approach and the other approaches in the literature in any article they publish. They have neglected a comparison with any other approaches in the literature.

Response: Following your suggestion, we have compared the overall yield and practical process of this work with what have been reported by our group and other groups for the synthesis of jet fuel range hydrocarbons with xylose (see Supplementary Tables 8 and 9 in the updated supporting

information). Some comments were also given at the page 11 of the revised manuscript.

I would also expect to see a Sankey diagram or a report of the overall yield of products. They need to discuss what happens to the water and NaCl that is in this process. I am on the fence about this paper but willing to consider a revision.

Response: We have added two Sankey diagrams and given the overall carbon yields of products from xylose and hemicellulose (see Supplementary Figs. 33 and 34 in the updated supporting information), respectively. Moreover, we also checked the reuseability of the NaCl aqueous solution (see the page 7 of revised manuscript and Supplementary Fig. 16 in the updated supporting information). It was found that the NaCl solution can be repeatedly used without any pretreatment, which is advantageous in real application.

Here are a couple of other minor comments about this article:

Comment 1. *The authors state that they have done recyclability tests of the catalysts. However all the recycling tests are done at high conversion where deactivation can be hidden. They should either characterize the catalyst after reaction to see if leaching of the Cu or change of Cu particle size occurred or they should do the stability tests at lower conversion where you can actually observe deactivation.*

Response: Thanks for your reminding. We have checked the recyclability of the Cu-La/SBA-15 catalyst at low xylose conversion. From the results illustrated in Figure R16, slight deactivation could be noticed during the repeated usage of the Cu-La/SBA-15 catalyst. However, such a problem can be overcome by the regeneration of the used Cu-La/SBA-15 catalyst by the calcination in flowing air at 773 K for 4 h and the reduction in 20% H₂/Ar flow at 773 K for 4 h. Following your suggestion, we also characterized the fresh and used Cu/SBA-15 and Cu-La/SBA-15 catalysts by ICP and TEM. From the results illustrated in Table R1 and Figure R17, it was found that the modification of Cu/SBA-15 by small amount of La significantly restrained the leaching of Cu species and the sintering of Cu particles, which may be one reason for the higher stability of the Cu-La/SBA-15 catalyst in the hydrogenolysis of xylose to CPO. We have added these results as Supplementary Table 1 and Supplementary Fig. 10 in the updated supporting information.

Figure R16. Xylose conversion and carbon yields of different products over the Cu-La/SBA-15 catalyst as the function of recycle time. Reaction conditions: $T = 438$ K, $P_{H_2} = 3$ MPa, 500 rpm, 1 h; 10 mL toluene, 10 mL 3 wt.% NaCl aqueous solution, 0.3 g xylose and 30 mg fresh Cu-La/SBA-15 catalysts were used for the test.

Table R1. Actual metal contents in the fresh and used Cu/SBA-15 or Cu-La/SBA-15 catalysts

Catalyst	Metal content measured by ICP-OES	
	Cu (wt.%)	La (wt.%)
Cu/SBA-15	23.1	—
Used Cu/SBA-15	16.6	—
Cu-La/SBA-15	24.5	3.7
Used Cu-La/SBA-15	22.9	3.7

Figure R17. TEM images of the fresh and used Cu/SBA-15 and Cu-La/SBA-15 catalysts.

Comment 2. The cartoon in Scheme 1 need to be improved and made more professional. My opinion (which could be wrong) are that the cartoons are not appropriate for a technical journal. I have no idea why a pill is running. The authors need to show what happens to the large amount of H₂O and NaCl that are used in this product. They also need to show any other byproducts.

Response: Sorry for this problem. We have improved the Scheme according to your suggestion (see Scheme R1 and Scheme 1 in the revised manuscript). We hope that you can be satisfied with it.

[Editorial Note: Scheme R1 redacted]

Scheme R1 Strategies for the selective production of methylindan and tetralin from xylose or the hemicellulose extracted from raw biomass.

Comment 3. What catalysts was used in Figure 1a? I assume it is Cu/SAB-15 as all the other captions state. The writing in the paper needs to be slightly improved to bring it up to journal standards. In Figures 3-5 it is very hard for me to tell the methylindene and naphthalene from the captions in the key. I suggest making those captions easier to read.

Response: Following your suggestion, we have added the catalyst information in the captions of Figure 1a and Figure 2a. Analogously, we have added the detail information of C₁₀ aromatics (including methylindan, methylindene, tetralin and naphthalene) in the captions of Figures 3-5.

Comment 4. The authors conclude with this statement, “A two-step method was developed for the manufacture of methylindan and tetralin from xylose or hemicellulose.” I think they are far from manufacturing these products in this route. Hemicellulose is a lot more complicated than just xylose. There are really no commercially available xylose streams on the market today. The authors need to show a process flow diagram with material and energy balances to state they could manufacture these products. The authors in this paper focus on the basic process chemistry which is an important first step but it is not correct to claim they have an actual process at this early stage.

Response: Sorry for this mistake. As you said, this manuscript focus on the basic process chemistry which is an important first step. It is still too early to talk about the manufacturing these products in this route. We have corrected it as “A two-step method was developed for the synthesis of methylindan and tetralin from xylose or hemicellulose”. Following your suggestion, we have also given a process flow diagram with material and energy balances (see Supplementary Fig. 32 and Supplementary Tables 5-7).

REVIEWERS' COMMENTS

Reviewer #2 (Remarks to the Author):

This revision has satisfactorily addressed the comments. I would recommend that this paper is accepted for publication in Nature Communications.

Reviewer #4 (Remarks to the Author):

The authors have responded to all of my concerns. I recommend the paper for publication. I think it will be a great contribution to the literature!